# Electronic Patient-Reported Outcomes in Nephrology: Focus on Hemodialysis

**DOI:** 10.3390/jcm11030861

**Published:** 2022-02-07

**Authors:** Rosa Pérez-Morales, Juan Manuel Buades-Fuster, Vicent Esteve-Simó, Manuel Macía-Heras, Carmen Mora-Fernández, Juan F. Navarro-González

**Affiliations:** 1Nephrology Service, Hospital Universitario Nuestra Señora de Candelaria, 38010 Santa Cruz de Tenerife, Spain; mmacia25@hotmail.com (M.M.-H.); jnavgon@gobiernodecanarias.org (J.F.N.-G.); 2Nephrology Service, Hospital Universitario Son Llàtzer, Carretera de Manacor, 07148 Palma, Spain; juanm.buades@gmail.com; 3Fundació Institut d’Investigació Sanitària Illes Balears, Carretera de Valldemossa, 79, Hospital Universitario Son Espases, Edificio S, 07120 Palma de Mallorca, Spain; 4Nephrology Service, Consorci Sanitari de Terrassa, Carretera Torrebonica, s/n, 08227 Terrassa, Spain; vesteve@cst.cat; 5Research Unit, Hospital Universitario Nuestra Señora de Candelaria, 38010 Santa Cruz de Tenerife, Spain; mmorfers@gobiernodecanarias.org; 6RICORS2040 (RD21/0005/0013), Instituto de Salud Carlos III, 28029 Madrid, Spain; 7Instituto de Tecnologías Biomédicas, Facultad de Ciencias de la Salud, Sección de Medicina, Universidad de La Laguna, 38200 Santa Cruz de Tenerife, Spain

**Keywords:** hemodialysis, ePRO, PROMs, PREMs

## Abstract

The success of hemodialysis (HD) treatments has been evaluated using objective measures of analytical parameters, or machine-measured parameters, despite having available validated instruments that assess patient perspective. There is an emerging interest regarding the use and relevance of patient-related outcomes (PROs). Electronic PROs (ePROs) involve the use of electronic technology, provide rapid access to this information, and are becoming more widely used in clinical trials and studies to evaluate efficacy and safety. Despite the scarce literature, this review suggests that ePROs are useful in providing a more customized and multidimensional approach to patient management and in making better clinical decisions in relevant aspects such as vascular access, duration and frequency of dialysis sessions, treatment of anemia, mental health, fatigue, and quality of life. The purpose of this review is to raise interest in the systematic use of ePROs in HD and to promote the development of studies in this field, which can respond to the gaps in knowledge and contribute to the implementation of the use of ePROs through new technologies, helping to improve the quality of health care.

## 1. Introduction

Chronic kidney disease (CKD) is a global health burden associated with significant morbidity, affecting around 10–16% of the general population worldwide, with a high impact on the quality of life of patients and their families, which is much higher when compared to other chronic diseases and cancer [1,2]. The success of treatments for CKD, and specifically for hemodialysis (HD), has been evaluated using objective measures of analytical parameters, or machine-measured parameters (dialysis monitor, blood pressure monitor, etc.) [1,3]. Despite the availability of validated instruments that measure patient perspective, their incorporation into clinical practice has been slow [3]. Nephrology care with an approach that focuses on getting to know the patient and family, considering their history, values, beliefs, priorities, preferences, symptoms, current situation, and future aspirations, is essential to ensure quality of care and to improve health outcomes [3]. In recent years, there has been increasing interest in the use and relevance of patient-reported outcomes (PROMs) and patient-reported experience measures (PREMs), collectively referred to as PROs (patient-related outcomes) [2,3,4]. Their routine and continuous use in nephrology could facilitate such a patient-centered approach to chronic kidney disease. The need to implement new technologies to improve the evaluation of nephrology care has been discussed for decades [5,6]. The SARS-CoV-2 pandemic has clearly impacted clinical activity in nephrology services worldwide, decreasing programmed activity and renal transplants. For this reason, a plan of care transformation with the implementation of telemedicine and the use of new technologies is necessary for the near future [7,8,9]. Electronic patient-reported outcomes (ePROs) encompass the use of digital technology to provide answers to standardized PRO questionnaires [3]. There is a knowledge gap on the optimal use of PROs in nephrology care, and there are few studies on the use, utility, and acceptability of ePROs in HD patients. The correct utilization of technology can facilitate the use of ePROs in hemodialysis and thus enhance patient-centered care. This review aims to raise interest among nephrologists in the routine use of ePROs and to promote studies in this field, which can provide answers to existing knowledge gaps and contribute to implementing their use, helping to improve the quality of care of HD patients.

## 2. What Are PROs (PROMs and PREMs), and Why Are They Important?

PROMs are self-report tools used to obtain assessments of health benefits, illness or medical treatment from the patient’s perspective in the form of a quality-of-life questionnaire or symptom questionnaire. In clinical practice, they have the potential to highlight relevant symptoms and symptom changes and to promote patient participation in their treatment. Their approach is strictly individualized and can only be used to improve outcomes specifically for that patient [3,4].

PREMs incorporate information about the patient’s experience of care as perceived by the patient. In routine clinical practice, they provide useful information about care management that can be used to improve the quality of clinical services in general, which will be of common benefit to all patients [2,3,4].

PROs (PROMs and PREMs), unlike anamneses, are provided directly by the patient and are not interpreted by healthcare professionals [3]. PROs allow us to determine whether our actions and treatment decisions improve the outcomes that matter most to patients and to enhance their experiences.

The routine uses in clinical practice of PRO data increase quality of care. However, they may have no impact on the process or outcomes of patient care, which may be related to the lack of knowledge about the appropriate use of PROs and their application in different settings, especially in nephrology. Since they pose a burden on the patient, if they are not used for decision making, they may lose the justification for performing them. This knowledge is necessary to develop strategies to guide the optimal use of PRO data [4].

In recent decades, drug regulatory authorities are paying more attention to PROMs data when making decisions on new drug approvals [10]. Even in the development of medical devices, the importance of PROMs is also beginning to be recognized, and evidence of this is that the United States Food and Drug Administration (FDA) has produced a document outlining a proposed conceptual framework to advance the development of rigorous and meaningful PROMs that can be used in clinical trials focused on the creation of innovative renal replacement therapy monitors [11].

## 3. What Type of PROs Exist and How Can They Be Used in Nephrology?

There are several generic PROMs that can be filled out by patients with a variety of diseases (SF-36, WHOQOL, WHOQOL-BREF, MQOLand, PHQ-9, etc.) and specific PROMs for renal diseases (KDQOL-SF, KDQOL-36, ESAS-r, KDQ, CHEQ, etc.) [3]. In nephrology, there is no consensus on which specific questionnaires should be used for routine evaluation of patients with advanced CKD. The Dialysis Symptom Index (DSI) has been considered the most relevant, comprehensive and simple symptom questionnaire [12]. Regarding the preference for recording PROMs data of renal patients in Europe, an expert consensus selected the Kidney Disease Quality of Life-36 (KDQOL™-36), as it shows both generic and disease-specific outcomes. Regarding PREMs instruments, there is no consensus for renal registries, as more studies are needed. It is recommended to include all patients on renal replacement therapy in the PROMs/PREMs questionnaire program, whose data should be collected at least once per year [2] (Table 1).

KDQOL-36 is the most widely used instrument in HD patients, both in its full version and in its 12-Item Short Form (SF-12). Although its reliability and validity has been demonstrated, one of its limitations is that it does not allow for the calculation of quality-adjusted life years (QALYs), which are used to evaluate the cost-effectiveness of health interventions. The instrument preferred by the National Institute of Health and Care Excellence (NICE) for estimating QALYs is the EQ-5D, which is not routinely used in dialysis, unlike the KDQOL-36. EQ-5D has five items (mobility, self-care usual activities, pain/discomfort and anxiety/depression) that measure health on the day of the survey with three (EQ-5D-3L) or five (EQ-5D-5L) descriptive levels for each item. A recent study has developed mapping algorithms based on mixture models to predict EQ-5D scores from generic KDQOL-36 [13]. These algorithms are already available and represent a new tool in the evaluation of health care actions and interventions for dialysis patients.

The need to further develop PROMs specific to renal disease subtypes is being prioritized in the US as well as other countries. PROMs should be defined in conjunction with objective markers of CKD severity or evolution to obtain a more comprehensive picture of clinical status and to enable effective reporting. The Patient-Centered Outcomes Research Institute published a series of standards to guide the future development of PROMs: asserting psychometric validity, minimizing patient burden, confirming significant change interpretation, sharing results with patients and clinicians, incorporating health information technology, and including patients with lower health literacy [19].

The use of routinely collected PROMs data in the real world also imposes several methodological challenges [20].

Many ideas have been suggested as to how PROs could be used in nephrology practice to support person-centered care, but there is a lack of robust evidence. Using PRO data at an individual level has been most explored both theoretically and empirically according to the literature and can be divided into: (1) patient use of PRO data to provide patient-centered care support, (2) patient use of their own PRO data to support patient engagement, (3) electronic collection of PROs to increase efficiency and more effectively support person-centered care and patient engagement, and (4) physician and patient use of individual-level PRO data to improve satisfaction, health, and quality of life outcomes [21].

Nair et al. [19] suggest a number of key considerations for successful implementation and adoption of PROMs in CKD, which would also include hemodialysis patients: (1) target patients who would particularly benefit from personalized care; (2) gather surveys online using a tablet or smartphone, completed either directly by the patient or by a nurse, (3) gather data on admission before an appointment, in dialysis centers during treatment, at home during home dialysis session, at home between appointments or dialysis sessions; (4) incorporate and store PROM data in the electronic medical record with password-protected access; (5) include them in risk prediction models, and compare them with national baselines; (6) share results with patients, caregivers and physicians; (7) adjust personalized medical treatments based on results (referral to psychologist, change in dialysis prescription, etc.).

## 4. What Are ePROs? What Are Their Advantages?

Electronic patient-reported outcomes systems (ePROs) encompass the use of electronic technology (such as computers, tablets, phones, apps) to provide responses to standardized instruments or PRO questionnaires [3]. They provide rapid access to this information for the healthcare team and are increasingly used in clinical trials and studies to evaluate the efficacy and safety of interventions from the patient’s perspective [22].

The use of ePROs instead of paper formats in clinical trials could improve the feasibility of PROMs assessment in routine clinical practice, as it eliminates the need for subsequent data entry and storage of questionnaires, as well as increasing the security of data protection. It makes data analysis and reporting easier by enabling data to be made available in exportable formats, with fewer errors and less missing data. It is more cost-effective in routine evaluation and has the potential for immediate scoring and presentation of results. It also has the potential to link PROMs to electronic medical record data, thus improving communication in multidisciplinary care and facilitating PROM assessment. [3,22,23]. Its widespread use has certain disadvantages, which should also be taken into account; the need to have an internet connection, a smartphone, computer or tablet, a certain degree of digital literacy or to have the support of a family member or healthcare personnel to carry out the digital survey in the event that the patient has a physical impediment or does not know how to deal with new technologies.

Physicians could actually use interactive ePROs devices to monitor and provide care to a large number of patients, while patients could access them through mobile devices to receive information about their health status and response to treatments in “real time”. As the responses are iterative, that is, the next question to appear depends on the previous response, it reduces the total number of responses and therefore the burden on the patient and their acceptance [10]. It also facilitates the use of this data at different healthcare levels: directly to the patient care department, extending to the level of healthcare facility management and administration, and even to the level of healthcare policy makers [3,20].

The use of ePROs has the potential to facilitate remote patient follow-up and improve efficiency by minimizing the need for hospital appointments, as well as improving patient outcomes such as quality of life and survival rates [10]. Patient and physician acceptability of routine collection of PROs in actual clinical practice is high. Despite this, the use of ePROs outcomes remains low [21].

## 5. What Evidence Do We Have on ePROs in Nephrology?

There is a lack of information on the use of ePROs in nephrology, especially in hemodialysis, although in recent years, interest in this field has increased.

A study conducted in 121 patients with stage 4 and 5 CKD (including patients on dialysis or after kidney transplantation), aimed at assessing the acceptability and practicality of the use of electronic data collection in patients with CKD, concluded that electronic data collection based to administer PROMs was acceptable and feasible for the majority of respondents, and therefore, it could be used to systematically assess PROMs among CKD patients both with and without replacement therapy. However, special attention must be paid to elderly patients with poor computer skills, as they may need additional assistance in completing the questionnaires [23].

The prospective non-interventional multicenter study PERCEPOLIS, which included 789 non-dialysis CKD patients, used for the first time a choice-based questionnaire design in a study on erythropoiesis stimulating agents (ESAs) in the elderly population. These data indicated that patients’ main expectations for ESAs were monthly injections and treatment efficacy. The efficacy of continuous erythropoietin receptor activator in maintaining stable hemoglobin within the recommended range was confirmed under real conditions. [24].

The Optimizing Participation in Routine Collection of Patient-Reported Electronic Outcomes (OPT-ePROs) study was intended to enable the implementation of ePROs within the context of secondary care for patients with chronic kidney disease in the United Kingdom. It involved a national infrastructure to securely collect, transfer and display data supplemented with materials and procedures to assist renal patients, including hemodialysis (HD) patients and healthcare staff with ePROs integrated into routine care pathways. This is the first study to provide a national ePROs data collection framework and, furthermore, propose a strategy to optimize the use of ePROs in these settings, addressing a gap in the literature [25].

A separate study on quality showed that provider feedback through real-time electronic health records (EHRs) is not sufficient to improve the acceptance of clinical practice guidelines and to change the therapeutic attitude in the care of patients with CKD, although they are considered necessary [26].

EHRs could provide a platform to incorporate PROs into clinical care in an effective and safe fashion. They provide more opportunities to act on PROs and thereby improve symptoms, providing relevant clinical information and decision support for healthcare staff, as well as self-management and peer community support for patients. A previous review demonstrated that focusing on measuring and monitoring PROs, such as pain and depression, can improve health-related quality of life (HRQoL) for patients with CKD [26].

Much of CKD management relies heavily on patient self-care, including medication and dietary adherence, self-monitoring blood pressure (BP), and daily physical activity. Evidence is growing showing that the inclusion of smartphone-based applications can support self-care in CKD and chronic diseases [27]. The development of ePROs applications for smartphones may be an important step in developing useful applications for both nephrologists and patients.

The number of healthcare providers developing ePROs systems has increased in recent years. Work is underway to facilitate ePROs user interfaces to reduce attrition rates in clinical trials and to improve adoption after implementation in clinical practice [10].

## 6. ePROs in Hemodialysis, an Unfinished Business

Few studies have been conducted focusing on the development of PROM questionnaires specifically for HD patients and to assess the usefulness and acceptability of ePROs.

A study conducted on HD patients in the US, describing the process and preliminary qualitative development of a new symptom-based PROM intended to assess physical symptoms related to HD treatment, laid the groundwork for the process of developing HD-specific PROMs. In this study, forty-two patients were interviewed for symptom-related concepts, and patient-reported concepts were used to generate a preliminary 13-item symptom PROM. Three rounds of cognitive interviews were then conducted with fifty-two patients to assess symptom relevance, item interpretability, and draft item structure, on the basis of an iterative refinement of the PROM. Responses and comments from participants during the cognitive interviews resulted in changes to the symptom descriptions, splitting the single item “nausea/vomiting” into two distinct items, removing the interference with daily activity items, and adding instructions, among others [28].

In the study by Schick-Makaroff et al. [22] involving ninety-nine patients on both peritoneal dialysis (PD) and home HD who completed via tablets two ePROs, the ESAS-r and the KDQOL™ -36, the data were used to discuss specific issues such as pruritus, appetite, insomnia, tiredness and dyspnea, as well as general health and the effects of CKD on daily life. Problems leading to a change in the care plan, referral to another professional or reassessment were pruritus, depression, fatigue, insomnia, anxiety and interference of the disease with daily life. The use of ePROs was found to be useful in the care of patients with home dialysis techniques. The same author also demonstrated that there is general satisfaction with the ePROs registry among patients receiving home hemodialysis [29].

Anemia is a worldwide complication of CKD patients on HD, and ESAs have been shown to improve clinical outcomes and quality of life. Under this premise, Staibano et al. [30] carried out a systematic review of 3533 studies published in Medline (Ovid), EM-BASE (Ovid), PsychINFO, and CINAHL databases. Of these, 67% were randomized controlled trials, 81% investigated patients with CKD, 14% patients after renal transplantation, and 5% evaluated patients on hemodialysis. The most common anemia intervention, utilized in 95% of the studies, was ESAs. Some 43% of the studies used a PROM not specific for CKD. Approximately one-third of the studies selectively reported PROM subscales, instead of reporting all subscales. Notable biases among the studies included lack of blinding, selective reporting of outcomes, and lack of power estimates of patient-centered outcomes (PCOs). No statistically significant association was found between improvements in hemoglobin and quality of life. Future studies using anemia and nephrology-specific PROMs should standardize methods of investigation and reporting of PCOs to achieve improved understanding of PCOs in HD.

The prevalence of depressive and anxiety symptoms in HD patients is high, and the routine use of PROMs can be used to assess patients’ mental health and care needs in this area. A study was conducted in Canada to describe the symptom burden of depression and anxiety reported by adults on HD and the perceptions of patients and nurses on this topic. Patient responses and notes from nurses’ electronic medical records related to mental health were collected. The mean age of the 408 patients included was 64 years, 57% were men and 87% were unemployed, 29% had depressive symptoms, 21% had anxiety symptoms, and 16% had both. It was concluded that PROMs (ESAS-r: Renal/EQ-5D-5L) had the potential to rapidly identify mental health problems. However, there were differing opinions on whether mental health fell within the scope of care of nephrologists and nurses, and there was consensus that more mental health resources were needed [31].

Fatigue is a prevalent and debilitating symptom in patients on HD therapy due to uremia, the treatment itself, and other comorbid conditions. It remains an under-recognized symptom, and the consequences are underestimated because it may not be visible in clinical settings. A systematic review was conducted by Jacobson et al. [32] with the aim of describing the experience of fatigue in patients undergoing chronic HD. Searches of MEDLINE, Embase, PsycINFO, CINAHL, reference lists, and PhD dissertations were conducted from baseline to October 2018, and sixty-five studies with 1713 HD patients were included. A total of four fatigue-related themes were identified: (1) debilitating and exhausting burden of dialysis (bodily exhaustion, post-dialysis burnout, vigilance and worry inhibiting rest, exhausting and agonizing regimen, and no remedy or relief); (2) restricted participation in life (time deprivation, management of energy reserves, need for rest and joy forfeited); (3) diminished ability to fulfill relationship roles (loss of ability to work and provide for family, failing as a parent, lack of stamina for sexual intimacy, and dependence on others); and (4) vulnerability to misunderstanding (being criticized for needing rest and not meeting expectations). From this review, it is clear that HD patients who experience fatigue are experiencing it as a relentless, severe exhaustion that permeates the entire body and limits the ability to perform usual activities and fulfill personal roles and aspirations. Explicit recognition of the impact of fatigue and the establishment of additional effective interventions to ameliorate fatigue is needed and can be achieved using PROMs in the clinical setting and in research.

Regarding quality of life and its association with the choice of conservative care (CC) on dialysis or without dialysis, a systematic review of eleven studies with 1718 patients was carried out comparing health-related quality of life (HRQoL) and symptoms. These are important findings for patients and physicians when deciding on the choice of preferred treatment. There were no randomized controlled trials, selection bias or confounding. In most studies, patients who opted for a CC were older and had more comorbidities and worse functional status than patients who opted for dialysis. Results were broadly consistent across studies, despite considerable clinical and methodological heterogeneity. Patient-reported physical health outcomes and symptoms were worse in patients who chose CC compared with pre-dialysis patients (before initial chronic dialysis treatment), but similar compared with patients on dialysis. Mental health outcomes were similar among patients who chose CC or dialysis, even before and after chronic dialysis initiation. In patients who opted for dialysis, the burden of kidney disease and impact on daily life increased after dialysis initiation. The available data, although heterogeneous, suggest that, in selected older patients, CC has the potential to achieve similar HRQoL and symptoms compared with the choice of dialysis. There is a need for high-quality prospective studies to confirm these results [33].

Patients spend a large percentage of time on dialysis (preparation, transfers, time in the waiting room and stay in the dialysis unit); thus, satisfaction with the care received is fundamental to their quality of life. In a study conducted in 103 HD and PD patients to explore the association between satisfaction with dialysis care (CHOICE, a PREM questionnaire) and quality of life (SF-36, a PROM questionnaire), a significant association was found between frequency of visits to the nephrologist and the physical component plus the mental component, information accuracy of the nephrologist and disease burden, accuracy of the instructions of the nephrologist and disease burden, coordination between the nephrologist and other physicians and mental component, focus on facility cleanliness and mental component, quantity of dialysis information available and burden of disease, information from staff regarding the choice between HD or PD and physical component plus burden of disease, and ability to see the social worker and burden of disease. The relationship between quality of life (with PROMs) and satisfaction care (with PREMs) is thus demonstrated, highlighting the central role of nephrologist–patient communication in the quality of life of dialysis patients [34].

The perceived importance to patients and their acceptability of longer and more frequent hemodialysis sessions had not previously been quantified. A choice experiment was conducted in which 183 hemodialysis patients were presented with a scenario consisting of twelve sets of treatment options followed by variable information on the clinical impact of the treatments offered. The described associations of improved survival and quality of life, reduction in the need for fluid restriction, and avoidance of additional access complications are strongly associated with the choice of longer or more frequent treatment regimens. Younger age, fatigue, previous experience of vascular access complications, non-heart failure, and shorter time spent making the trip to dialysis centers were associated with a preference for four weekly sessions. Patients were willing to trade up to 2 years of life to avoid 4 weekly session regimens or vascular access complications. Upon application of the estimated benefits and harms of treatment from the existing literature, the fully adjusted model revealed that 27.1% would choose longer regimens administered three times per week, and 34.3% would choose 4 hours four times per week. Analogous estimates for younger fatigued patients living near their unit were 23.5% and 62.5%, respectively. The anticipated acceptance of longer and more frequent HD regimens exceeds their use in current clinical practice. These findings underscore the need for robust data on the clinical efficacy of these more intensive regimens and broader consideration of patient choice in the choice of dialysis regimens [35].

In terms of vascular access, using central venous catheters for vascular access in HD patients has been associated with a higher risk of complications in comparison with arteriovenous fistulas (AVF). Nevertheless, catheter use remains high, and patient satisfaction could be an important factor in its use. The Vascular Access Questionnaire (VAQ) was designed to measure patients’ ratings of vascular access. In total, 227 CKD patients on HD from two centers were asked to rate how much they were bothered by 17 vascular access-related problems. VAQ symptom scores were then compared among patients using catheters and those using fistulas for vascular access. Symptom scores did not differ between patients using catheters and those using fistulas. Older patients had lower symptom scores with catheters than with fistulas. Patients seem to be mainly concerned about the appearance of access and complications related to cannulation, especially the elderly. Further education on the risk of adverse events with catheters and implementation of measures aimed at reducing cannulation-related complications may help to increase fistula rates and improve patient satisfaction with their vascular access [36].

Despite the scarce literature on PROs applied to HD, it is useful to provide a more personalized and multidimensional approach to patient treatment and to make better clinical decisions (Table 2).

## 7. What Is the Future of ePROs in HD?

The International Consortium for Health Outcomes Measurement (ICHOM) [37] is leading a series of initiatives aimed at measuring and communicating patient health outcomes in a standardized way, being an example to follow in order to advance toward the achievement of this model. ICHOM is a non-profit organization whose aim is to facilitate the standardized measurement of health outcome variables, integrating the perspective of healthcare professionals and patients. In this way, it aims to achieve “value-based healthcare”, high quality and optimal patient outcomes.

To date, ICHOM has developed twenty-eight standardized sets of variables, including CKD, which have been implemented in more than six hundred institutions in numerous countries and patient registries at a national level [37]. All sets developed by this consortium include two categories of health variables: (1) case-mix variables: these make it possible to characterize the patient and therefore contextualize the results according to the patient; (2) clinical outcomes and PROs collected at baseline and during follow-up, making it possible to determine the evolution of the disease indicators. Two categories of follow-up variables were established: essential and optional.

The set, developed by ICHOM for the follow-up of patients with CKD, is aimed at patients in advanced stages with a high risk of CKD progression (G3a/A3 and G3b/A2-G5), and patients requiring renal replacement therapy with dialysis, transplantation or conservative treatment [38]. The variables were categorized into four groups (survival, disease burden, PROs, and specific outcomes-modality of treatment). Experts from different fields and nationalities participated in the development of the set of ICHOM variables: patient representatives, specialized nurses, registry experts, surgeons, and others. Variables on survival, disease burden (hospitalizations and cardiovascular events), specific results of each of the renal treatment options (specifically in hemodialysis, survival of vascular access and residual renal function) and PROs were differentiated.

The PROs are common to all CKD patients and therefore to hemodialysis patients. They assess quality of life, pain, physical function (daily activities and fatigue), which are precisely those that were most highly rated by the group of patient representatives.

In Spain, the “Cercano” project is being developed [39], which aims to adapt the standardized set of variables developed by ICHOM to the Spanish Health System for subsequent implementation. The work is being carried out by the Clinical Management Group of the Spanish Society of Nephrology. The instruments to be used to assess PROMs will be PROMIS in its ePRO version and SF-36 and EuroQoL 5D-5 used alternatively.

## 8. Conclusions

Although still with limited experience, it has been proven that the use of ePROs is useful in the care of hemodialysis patients related to different aspects, including quality of life and their relationship with different potential problems and complications such as anemia, vascular access, and individualization of dialysis parameters or chronic fatigue, as well as aspects related to mental and psychological health such as anxiety or depression. In addition, these tools will make it possible to assess aspects related to patient satisfaction and preferences, which are crucial factors for a holistic optimization of dialysis therapy.

The incorporation of ePROs into clinical practice will have the potential to provide deep insight into a person’s disease experience, make clinical trials more effective, transform initiatives into health policy, and individualize high-quality care for patients with chronic kidney disease, especially on hemodialysis.

## Figures and Tables

**Table 1 jcm-11-00861-t001:** Main current patient-reported outcomes measures developed for adults with chronic kidney disease (CKD).

STUDY Ref.	Assessments	Item Number	Burden Rating Scale	Population/Validation	Recall
CKD-SBI [14]	Prevalence, severity and frequency of symptoms	33	11 point Likert scale	CKD/ESRD	4 weeks
CHEQ [15]	Health perception, physical, social, physical role, emotional role, pain, mental compound, vitality, cognitive and sexual disorder, sleep, job, recreation, travel, finances, general QoL, diet, body image, dialysis access, symptoms	80	2–7 point Likert scale	ESRD/CKD	4 weeks/3 months/in general
DSI [16]	Physical symptom burden, symptom severity	30	5 point Likert scale	ESRD/CKD	1 week
KDQOL-SF [17]	Symptoms, burden of kidney disease, work situation, cognitive impairement, social aspects, sexual disorder, sleep, social support, patient satisfaction, physical functioning, role physical, pain, general health perceptions, emotional well-being, emotional state, social function, energy	82	2–10 point Likert scale	ESRD/CKD	4 weeks
KDQOL-36 [18]	Includes the SF-12 as generic core plus the burden, symptoms/problems, and effects of kidney disease scales from the KDQOL-SF™v1.3.	36	5 point Likert scale	ESRD/CKD	4 weeks/in general

CKD: chronic kidney disease; CKD-SBI: Chronic Kidney Disease-Symptom Burden Index; CHEQ: CHOICE Health Experience Questionnaire; CKD QOL: Chronic Kidney Disease Quality of Life; DSI: Dialysis Symptom Index: KDQOL-SF: Kidney Disease Quality of Life—Short Form: KDQOL-36: Kidney Disease Quality of Life-36. References [14,15,16,17,18] are included.

**Table 2 jcm-11-00861-t002:** Summary of the studies included in the review on the evidence of PROM in hemodialysis and its contribution.

Author ^Ref^	Year	Contribution
Flythe et al.	2019	Laid the foundations of the methodology for developing dialysis-specific PROM questionnaires.
Schick-Makaroff, K. et al.	2019	Proved that the use of ePROs is useful in home dialysis techniques.
Staibano, P. et al.	2020	Proposed the standardization of research methods and the reporting of PROMs in HD.
Schick-Makaroff, K. et al.	2017	Demonstrated that there is general satisfaction with the ePROs registry among patients receiving HD at home.
Schick-Makaroff, K. et al.	2021	Suggested that PROM questionnaires (ESAS-r: Renal/EQ-5D-5L) can quickly identify mental health problems.
Jacobson, J. et al.	2019	Proved that PROMs in clinical and research settings can improve the detection and treatment of fatigue in HD.
Verberne, W.R. et al.	2021	Advanced that the use of PROMs in selected patients has the potential to reach a similar QoL in patients on CC or dialysis.
Cirillo, L. et al.	2021	Proved the relationship between satisfaction with care and QoL, highlighting the central role of nephrologist-patient communication in the QoL of dialysis patients.
Fotheringham, J. et al.	2021	Demonstrated the importance of the patient preferences in the selection of more frequent or longer HD or regimens.
Quinn, R.R. et al.	2008	Proved that information on catheter and fistula care decreases the number of complications and increase patient satisfaction with their vascular access.

CC: conservative care; HRQoL: health-related quality of life; QoL: quality of life, [22,28,29,30,31,32,33,34,35,36].

## Data Availability

This study does not report any direct data.

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
