# Peer review of "Electronic Patient-Reported Outcomes in Nephrology: Focus on Hemodialysis"

_jcm, 2022, doi:10.3390/jcm11030861_

Round 1
Reviewer 1 Report
In my opinion the paper is of utmost importance but some parts should be extensively rewritten, including language editing.
- Chronic kidney disease (CKD) is a pathology with significant associated morbidity, 38 and although it affects a small percentage of the general population
According to some data the prevalence of CKD in Western countries may reach up to 7-10%, so I would disagree with above cited sentence.
- The introduction should be more detailed and provide the aim of the paper.
- Definitions of PREM and PROM should be described more precisely with focus on differences between them.
- The conclusions are in my opinion too general. This part should shortly summarize the most important issues of the review and highline those of utmost interest.
Author Response
First of all, we would like to thank the reviewer for his suggestions and comments. The manuscript has been modified which has improved its quality.
According to the reviewer’s suggestions, some parts of the manuscript have been extensively revised and rewritten, including language editing.
- Data on chronic kidney disease (CKD) relevance and prevalence has been updated. Nowadays, CKD is a global health burden with a prevalence around 10-16% in the general population. This has been included in the fists sentence of the Introduction.
- The introduction has been rewritten to include more detailed information as well as the aim of the paper.
- Section 2 has been also revised and rewritten. The title has been change to include PROMs and PREMs, and the text has been modified to set out more precisely and in more detail the definitions of PREM and PROM, highlighting the differences between them.
- The conclusions have been modified to briefly highlight the positive results with the use of ePROs and the potential benefits of its incorporation as a routine tool into clinical practice in the context of hemodialysis.
Reviewer 2 Report
In the manuscript “Electronic patient Reprted Outcomes in Nephrology: Focus on Hemodialysis” by Perez-Morales R et al. The authors analyze an electronic tool (ePRO) and its implementation in clinical practice to improve the quality of patent care in HD patients.
This is a theoretical explanation of the benefits of introducing new technology and how their application can help as an introduction to understanding and applying it in everyday clinical practice.
The paper is written on 13 pages and consists of 8 parts: 1) introduction, 2) What Are PROs and Why Are They Important, 3) What Type of PROs Exist and How Can They Be Used in Nephrology, 4) What Are PROs ? Are Their Advantages?, 5) What Evidence Do We have on e PROs in Nephrology?, 6) ePROs in Hemodialysis and Unfinished Business, 7) What Is the Future of e PROs in HD ?, 8) Conclusion.
The manuscript is clear, relevant for the field and presented in a well-structured manner.
37 references have been cited that have been published in the last 5 years.
Although the authors are very happy about the application of telemedicine in nephrology, this paper lacks comment on the shortcomings related to the use of ePROs.
I recommended that the paper be accepted after minor revision.
Author Response
First of all, we would like to thank the reviewer for his suggestions and comments. The manuscript has been modified, which has improved its quality.
According to the reviewer’s suggestion, additional information about shortcomings related to the use of ePROs has been included in the manuscript (section 4, lines 14-18, and section 6, eighth paragraph.
Round 2
Reviewer 1 Report
I have no further remarks.
Author Response
Thank you very much for your indication.
The text has been spell-checked and the errors have been corrected.